# Masked Snake Attention for Fundus Image Restoration with Vessel Preservation

Submission Id: 1651

## ABSTRACT

Restoring low-quality fundus images, especially the recovery of vessel structures, is crucial for clinical observation and diagnosis. Existing state-of-the-art methods use standard convolution and window based self-attention block to recover low-quality fundus images, but these feature capturing approaches do not effectively match the slender and tortuous structure of retinal vessels. Therefore, these methods struggle to accurately restore vessel structures. To overcome this challenge, we propose a novel low-quality fundus image restoration method called Masked Snake Attention Network (MSANet). It is designed specifically for accurately restoring vessel structures. Specifically, we introduce the Snake Attention module (SA) to adaptively aggregate vessel features based on the morphological structure of the vessels. Due to the small proportion of vessel pixels in the image, we further present the Masked Snake Attention module (MSA) to more efficiently capture vessel features. MSA enhances vessel features by constraining snake attention within regions predicted by segmentation methods. Extensive experimental results demonstrate that our MSANet outperforms the state-of-the-art methods in enhancement evaluation and downstream segmentation tasks.

## CCS CONCEPTS

• **Computing methodologies** → **Reconstruction**.

## KEYWORDS

Fundus Image Restoration, Vessel Preservation, Attention

## 1 INTRODUCTION

Retinal vessel characteristics, such as vessel diameter, branching angles, and branching lengths, are important biomarkers for many retinal and systemic diseases, including diabetic retinopathy [34], glaucoma [23], macular degeneration [37], hypertension [15], and cardiovascular diseases [10]. Unfortunately, there are many factors that lead to the degradation of fundus image quality, such as imperfections in the fundus camera optics, improper camera adjustment, or defocusing during the exam. Low-quality fundus images result in unclear retinal vessel structures, hindering reliable diagnosis by ophthalmologists and affecting automated image analysis. The

*ACM MM, 2024, Melbourne, Australia*
© 2024 Copyright held by the owner/author(s). Publication rights licensed to ACM.
ACM ISBN 978-x-xxxx-xxxx-x/YY/MM
https://doi.org/10.1145/nnnnnnn.nnnnnnn

performance of vessel semantic segmentation collapses greatly on low-quality fundus images, as shown in Fig. 1(a).

Recently, many deep learning-based methods [7, 26, 31] for low-quality image restoration have been proposed, achieving good results in natural scenes. However, fundus images contain specific anatomical structures such as vessels, these methods treat each pixel equally important without restoring the key retinal vessel structures, making them unsuitable for low-quality fundus image restoration tasks. Some methods [4, 6, 22] based on GANs [11] utilize discriminators to align the restored images with high-quality images. However, these methods often restore images with a similar style to high-quality images, making it challenging to accurately restore vessels. To guide the network in preserving retinal vessel structure, existing methods [12, 17–20, 29, 36] utilize kinds of priors to constrain retinal structures in the enhancement model. Some methods [12, 29, 36] such as CofeNet [29] introduced pretrained ResNet-34 [14] to provide semantic information, but these approaches come with additional training time and data cost. To avoid these costs, other methods such as PCE-Net [20] use Laplacian Pyramid [1] to provide the structure information. They directly concatenate prior information at the feature level of the enhancement model in a data-agnostic manner, disregarding the fine, long, and continuous morphological characteristics of the vessels. This oversight results in the full potential of the prior information not being fully realized.

Ideally, one would expect to enhance retinal vessel features by selectively attending to more informative regions on a data-dependent basis under the guidance of prior information and domain knowledge. Vessel is a kind of thin and tortuous tubular structures, and accounts for only a small proportion of the overall image with limited pixel composition. In response to this specificity of tubular structures, DSCNet [25] proposes a dynamic snake convolution to achieve accurate segmentation of tubular structures by more flexibly focusing on slender and tortuous local structures with the help of deformable offset [5, 39]. But dynamic snake convolution [25] lacks the element relation modeling mechanism. This motivates us to explore a flexible attention pattern in low-quality fundus image restoration tasks to assist the network in adaptively enhancing vessel features under the guidance of domain knowledge of vessel, thus better preserving vessel structures.

In such context, we propose a novel approach named Masked Snake Attention Network (MSANet), as shown in Fig. 2, towards a fundus image restoration model that can accurately preserve retinal vessel structures. The core of our MSANet is a novel Masked Snake Attention module(MSA), which adaptively aggregates vessel features in the regions indicated by the prior information (as shown in Fig. 4). To be specific, we first use traditional segmentation method Optimally Oriented Flux (OOF) [16] on low-quality images to obtain vessel segmentation masks as prior information without training cost. Then, because vessels account for only a small proportion of

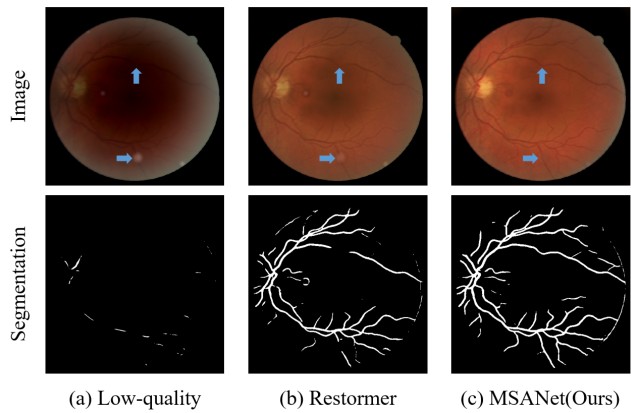

| (a) Low-quality | (b) Restormer | (c) MSANet(Ours) |

**Figure 1: Three kinds of fundus images and their retinal vessel segmentation results on DRIVE [30]. (a) Low-quality fundus image, (b) Restored fundus image by Restormer [38], (c) Restored fundus image by our MSANet. All of the segmentation results are applied by a pretrained U-Net [27] on corresponding fundus images.**

the overall image with limited pixel composition, we apply snake attention on localized features centered around predicted segments by a predefined threshold. This allows the network to capture vessel features under the guidance of prior information. In the intermediate layers of the network's decoder, we apply Snake Attention module (SA) to adaptively aggregate vessel features based on the morphological structure of the vessels without any semantic prior information. Specifically, we employ an iterative strategy and offsets to flexibly select the continuous key/value for a given query under the continuous constrain, thereby enhancing the network's capability of modeling vessel geometric transformations. The main contributions of this paper are summarized as follows:

(1) We design a noval method called Masked Snake Attention Network (MSANet) specifically to address the issue of retaining retinal vessel preservation in low-quality fundus image restoration tasks.

(2) Taking into account the thin, long, and tortuous shape characteristic of vessels, we design a novel feature aggregation module called Snake Attention to adaptively aggregate vessel features based on the morphological structure of the vessels. Guided by semantic prior information provided by traditional semantic segmentation method, we further propose Masked Snake Attention to restrict the snake attention on localized features centered around predicted segments.

(3) Experimental results demonstrate that our proposed method achieves superior performance compared to state-of-the-art approaches in enhancing low-quality fundus images and downstream segmentation tasks.

## 2 RELATED WORK

### 2.1 Fundus Image Restoration

Fundus image restoration methods have two main categories: hand-crafted methods and deep learning based methods.

**Hand-crafted methods.** Traditional fundus image enhancement methods are mainly based on hand-crafted priors. Contrast enhancement can widen the dynamic ranges of images, thereby improving image readability. Therefore, Contrast Limited Adaptive Histogram Equalization (CLAHE) is a popular method for restoring degraded fundus images. For instance, Setiawan et al. [28] applied CLAHE in the green channel to enhance the quality of color retinal images. Instead of simply considering the color and texture information, some approaches [9, 24, 33] decompose the reflection and illumination, achieving image enhancement and correction by estimating the solution in an alternate minimization scheme. Moreover, SGRIF[3] is a method based on Guided Image Filtering (GIF) [13], which can improve the contrast of the image and maintain the edges when restoring cataract-affected fundus images. These hand-crafted prior-based methods heavily rely on global image statistics, leading to their sensitivity to data and unsatisfactory performance on low-quality fundus images in real-world scenarios.

**Deep learning based methods.** Recently, deep learning has become the mainstream method in the field of computer vision due to its superiority in image representation. Deep learning-based methods have achieved satisfactory performance in many image restoration tasks such as image denoising [26], image dehazing [7], and low-light image enhancement [31]. However, due to the rich retinal vessel characteristics present in fundus images, which are critical for disease diagnosis, these deep learning methods designed for natural scenes are not suitable for fundus image restoration. Some methods [4, 6, 22] designed within the Generative Adversarial Network (GAN) [11] framework attempt to learn a suitable mapping from a low-quality domain to a high-quality domain. However, these GAN-based methods tend to generate images that are similar in style to high-quality images without effectively preserving the important details of vessels. To address this issue, many methods [12, 29, 36] have introduced semantic priors. For example, Cofe-Net [29] is designed to preserve the retinal structures in the low-quality fundus image restoration process by introducing a semantic segmentation network for vessels. However, pretraining a semantic segmentation network requires a certain amount of annotated data and training time cost. Other methods [17–20] utilize structural priors such as high-frequency features, thus avoiding additional training and annotation costs. These prior-based methods [12, 17–20, 29, 36] typically directly concatenate prior information into the features of the enhancement model, limiting the potential of the prior information. In contrast, we introduce a flexible module named Masked Snake Attention to explore the potential of priors and thus produce visually pleasing enhanced results.

### 2.2 Deformable CNN and attention

Deformable convolution [5, 39] is a powerful mechanism to attend to flexible spatial locations conditioned on input data. Recently, it has been applied to design various attention mechanisms [32, 35, 40]. Deformable DETR [40] incorporates the sparse spatial sampling of deformable convolution, allowing it to achieve better performance than DETR[2] with fewer iterations. DAT [35] motivated by Deformable convolution [5, 39] presents a simple and efficient

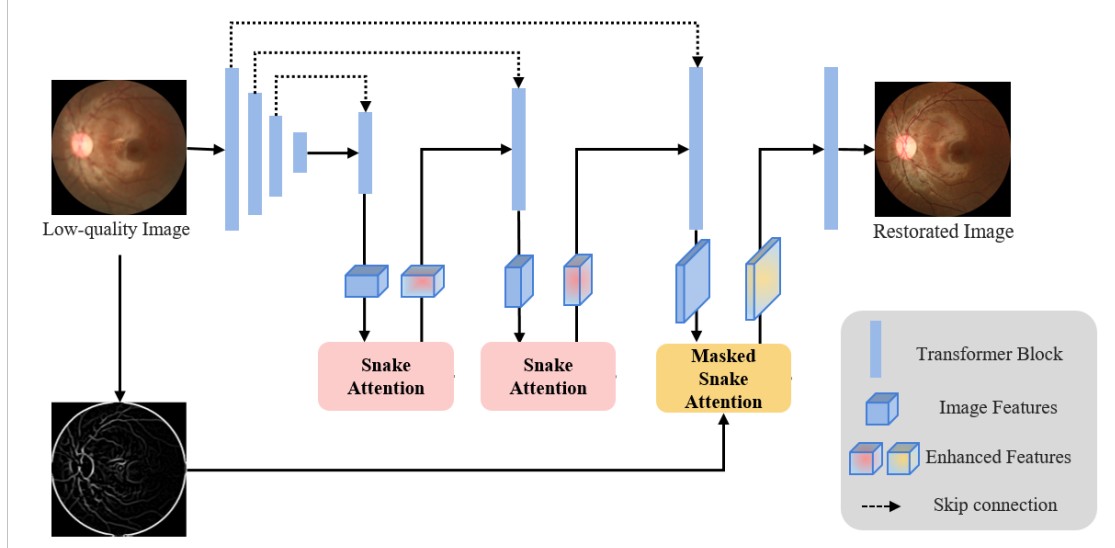

**Figure 2: Pipeline of our Masked Snake Attention Network (MSANet).**

deformable self-attention module which is used for designing a powerful backbone for various vision tasks. Wang et al. [32] borrowed the sampling strategy of Deformable DETR [40] and extended it to the video domain by leveraging motion cues stored in compressed video to identify the most salient patches in frames different from the query patch. These methods designed based on Deformable convolution [5, 40] have achieved satisfactory performance across multiple applications, fully demonstrating their ability to focus on relevant regions in a data-dependent manner and capture more informative features in natural scenes. However, vessels are thin, long, and curved structures, making it difficult for these methods based on deformable convolution to efficiently focus on the thin tubular structures. Qi et al. [25] considered the snake-like morphology of tubular structures and designed a Snake convolution, which can stably enhance the perception of tubular structures in the feature extraction process. This inspires us to design a deformable attention module for low-quality fundus image restoration tasks to assist the network in adaptively enhancing vessel features under the guidance of prior information, thereby better preserving vessel structures.

## 3 METHODOLOGY

In this section, we present in detail Masked Snake Attention(MSANet), which consists of Snake Attention module and Masked Snake Attention module, as illustrated in Fig. 2. Details of each module are discussed in the following.

### 3.1 Motivation and Overview

As depicted in Fig.2, MSANet is constructed upon Restormer [38], which is a U-Net-like Transformer model. The encoder comprises four layers of Transformer blocks, while the decoder comprises three layers of Transformer blocks. To assist the recovery process, the encoder features are concatenated with the decoder features

via skip connections [27]. In the decoder section, we introduce one Masked Snake Attention module (MSA) and two Snake Attention modules (SA) to accurately preserve thin, long, and continuous vessel structures. Both MSA and SA select key and value positions in a data-dependent manner to adaptively aggregate vessel features based on the morphological structure of the vessels. Because priors can provide a wealth of information for enhancing performance, we introduce MSA at the top layer of the decoder to enhance vessel features by constraining snake attention within regions predicted by the Optimal Oriented Flux (OOF) [16] mask. Therefore, MSA and SA can adaptively aggregate features in a manner that fits the vessel structure, thus helping the network preserve vessel structures.

### 3.2 Snake Attention

Dynamic snake convolution [25] can improve network's ability to adapt to the geometric variations of tubular structures by offsetting the grid sampling locations of standard convolution with displacements learned with respect to the preceding feature maps. Prior work in [25] has shown the effectiveness of dynamic snake convolution in segmenting tubular structures in various seniors. However, dynamic snake convolution lacks the element relation modeling mechanism. To deal with it, we propose Snake Attention module (SA), which combines the advantage of the sparse spatial sampling of dynamic snake convolution and the relation modeling capability of Transformers. In this section, we discuss how to perform SA to flexibly extract the local vessel features.

Given the standard window-based self-attention block is formulated as:

$$q = xW_q, k = xW_k, v = xW_v, \qquad (1)$$

$$z = \sigma\left(qk^\top / \sqrt{d}\right) v \qquad (2)$$

$$z = zW_o \qquad (3)$$

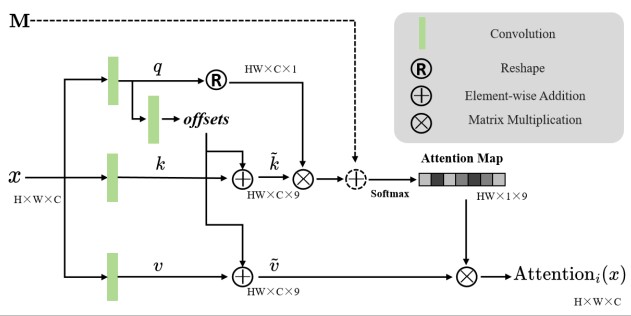

Figure 3: Architecture of the Snake Attention module (SA) from x-axis and y-axis directions.

where $\sigma(\cdot)$ denotes the softmax function, and $d = C$ is the dimension of $x$. $z$ denotes the embedding output, $q, k, v \in \mathbb{R}^{N \times C}$ denotes query, key, value embeddings respectively. $W_q, W_k, W_v, W_o \in \mathbb{R}^{C \times C}$ are projection matrices.

To enable the network to focus on complex geometric features of the target, inspired by [5, 39], we introduce deformable offsets. As illustrated in Fig. 4 the input feature map $x \in \mathbb{R}^{H \times W \times C}$ is firstly projected linearly to obtain the query tokens $q = xW_q$, where $H$, $W$ and $C$ are respectively the height, width and channel of the feature map in decoder. And then $q$ is fed into a light weight sub-network $\theta_{offset}(\cdot)$ to generate the offsets $\Delta p = \theta_{offset}(q)$. Additionally, we constrain the offsets within a certain range, such as $[-1, 1]$. However, considering the unique geometric characteristics of vessels, the deformation offsets without any additional constraints can easily cause the deformed features to stray outside the vessel. Inspired by [25], we adopt an iterative strategy to select the subsequent position to be observed in turn for each target to be processed, ensuring the continuity of deformed feature. Then the features are sampled at the locations of deformed points as keys and values, followed by projection matrices:

$$q = xW_q, \widetilde{k} = \widetilde{x}W_k, \widetilde{v} = \widetilde{x}W_v, \tag{4}$$

$$\text{with } \Delta p = \theta_{offset}(q), \widetilde{x} = \phi\left(x; p + \sum \Delta p\right). \tag{5}$$

where $\widetilde{x}$ is deformed from $x$, $\widetilde{k}$ and $\widetilde{v}$ represent the deformed key and value embeddings respectively. As the offset $\Delta p$ is typically fractional, we set the sampling function $\phi(\cdot, \cdot)$ to a bilinear interpolation to make it differentiable:

$$\phi(x; p) = \sum_{(r_x, r_y)} G(p, r)x(r) \tag{6}$$

where $p$ denotes a fractional location, $r$ enumerates all integral spatial locations, and $G(\cdot; \cdot)$ is the bilinear interpolation kernel and it is separated into two one-dimensional kernels as:

$$G(p, r) = g(p_x, r_x)g(p_y, r_y) \tag{7}$$

where $g(a, b) = max(0, 1 - |a - b|)$.

We get $\widetilde{x}$ along the x-axis and y-axis respectively as shown in Fig. 3. Taking the x-axis direction as example, the specific position of each pixel in $p^{i \pm c}$ is represented as: $p^{i \pm c} = (p_x^{i \pm c}, p_y^{i \pm c})$, where $c = \{0, 1, 2, 3, 4\}$ denotes the horizontal distance from the central

pixel $p^i = (p_x^i, p_y^i)$. The whole selection of each $p^{i \pm c}$ is a cumulative process, which means each $p^{i \pm c}$ is determined by the previous $p^{i \pm (c-1)}$. Specifically, we add vertical offsets together to ensure the continuity of the deformed point. The formulation for deformed points $p^{i \pm c}$ along the x-axis is as follows:

$$p^{i \pm c} = \begin{cases} (p_x^{i+c}, p_y^{i+c}) = (p_x^i + c, p_y^i + \sum_i^{i+c} \triangle p_y) \\ (p_x^{i-c}, p_y^{i-c}) = (p_x^i - c, p_y^i + \sum_{i-c}^i \triangle p_y) \end{cases} \tag{8}$$

The formulation for deformed points along the y-axis is as follows:

$$p^{j \pm c} = \begin{cases} (p_x^{j+c}, p_y^{j+c}) = (p_x^i + \sum_i^{i+c} \triangle p_x, p_y^i + c) \\ (p_x^{j-c}, p_y^{j-c}) = (p_x^j + \sum_{i-c}^i \triangle p_x, p_y^j - c) \end{cases} \tag{9}$$

We perform attention on $q$, $\widetilde{k}$ and $\widetilde{v}$ along the x-axis and y-axis. The output of snake attention is formulated as:

$$\text{Attention}(x) = \sigma\left(q\widetilde{k}^\top / \sqrt{d}\right)\widetilde{v} \tag{10}$$

$$z = \text{Attention}_i(x) + \text{Attention}_j(x) \tag{11}$$

$z$ is projected through $W_o$ to get the final output $z$ as Eq. 3. As shown in Fig. 4 and Fig. 5, our snake attention covers a $9 \times 9$ range during the deformation process due to the two-dimensional (x-axis, y-axis) changes.

## 3.3 Masked Snake Attention

Semantic priors can provide a wealth of information for improving the enhancement performance, we design the Masked Snake Attention module(MSA) to apply snake attention on masked regions as shown in Fig. 5. In our method, we choose traditional semantic segmentation method Optimal Oriented Flux (OOF) [16] to provide semantic prior information without annotated data and training time cost. Specifically, as illustrated in Fig. 2, we use the OOF masks predicted on low-quality fundus images to determine the regions for snake attention based on a predefined threshold.

We define $\mathbf{M} \in [0, 1]^{H \times W}$ is the vessel segmentation mask predicted by OOF [16]. We can obtain the attention mask $\mathcal{M} \in \{0, 1\}^{H \times W}$ based on the predefined threshold(q-th quantiles of $\mathbf{M}$ is uesed as $threshold$, where $q = 90\%$) at query features location $(r_x, r_y)$:

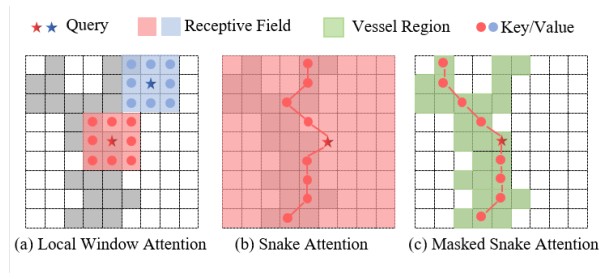

**Figure 5: Comparison of Masked Snake Attention, Snake Attention and Local Window Attention. The red and blue stars note the query, the red and blue squares note corresponding receptive regions, and the green squares denote the vessel regions which the queries in Masked Snake Attention attend. (a)Local Window Attention aim to extract local features. (b)Snake Attention can capture features adaptively based on vessel structures. (c)Masked Snake Attention can more accurately capture vessel features under the guidance of mask than Snake Attention.**

$$\mathcal{M}(r_x, r_y) = \begin{cases} 0 & \text{if } \mathbf{M}(r_x, r_y) \geq threshold \\ -\infty & \text{otherwise} \end{cases} \quad (12)$$

According to Eq. 10, our Masked Snake Attention modulates the attention matrix via

$$\text{Attention}(x) = \sigma\left(q\widetilde{k}^\top / \sqrt{d} + \mathcal{M}\right)\widetilde{v} \quad (13)$$

## 4 EXPERIMENTS

### 4.1 Datasets

The experiments used three datasets:

EyeQ [8]: a public dataset based on the Kaggle Diabetic Retinopathy Detection dataset [32] contains 28,792 samples (16,817 'Good', 6,435 'Usable', and 5,540 'Reject') sorted according to image quality. We follow the degradation pipeline described in [29] to synthesize corresponding low-quality images based on images labeled as 'Good' as training dataset.

Real Fundus (RF) [6]: a public dataset collected by Shenzhen Eye Hospital consists of 120 LQ and HQ clinical fundus image pairs captured by ophthalmologists using a ZEISS VISUCAM200 fundus camera. We choose the whole dataset as our testing dataset for full reference enhancement evaluation.

DRIVE [30]: a retinal vessel segmentation dataset consisting of 40 color retinal images with segmentation masks. We only use 20 images from test dataset for segmentation evaluation. Follow [29], the images from DRIVE [30] are used to generate 100 low-quality images for quantitative assessment.

### 4.2 Implementation Details

Our method is implemented by PyTorch, and trained on a single NVIDIA RTX 3090 GPU. We train models with AdamW optimizer ($\beta1 = 0.9$, $\beta2 = 0.999$, weight decay $1e^{-4}$) and L1 loss for 300K iterations with the initial learning rate $1e^{-4}$ gradually reduced to

**Table 1: Quantitative evaluation of MSANet compared with different methods.**

| Method | Full reference | | Segmentation | |
|---|---|---|---|---|
| | PSNR | SSIM | IoU | Dice |
| SGRIF[3] | 20.9950 | 0.6855 | 0.4513 | 0.6096 |
| StillGAN[22] | 24.7611 | 0.7393 | 0.4970 | 0.6582 |
| Cofe-Net[29] | 18.0041 | 0.7532 | 0.5121 | 0.6748 |
| I-SECRET[4] | 24.7536 | 0.7757 | 0.5098 | 0.6731 |
| PCE-Net[20] | 19.5231 | 0.7240 | 0.5346 | 0.6946 |
| RFormer[6] | 24.9654 | 0.6277 | 0.5434 | 0.7022 |
| Ours | **25.7261** | **0.8068** | **0.5477** | **0.7064** |

$1e^{-6}$ with the cosine annealing [21]. All of the images are re-scaled to the size of $512 \times 512$ for fair comparison with other methods. The mini-batch size is set to 2. During the training procedure, fundus images are first cropped into the patches with the size of $128 \times 128$. Then the patches are fed into the model. For data augmentation, we use horizontal and vertical flips.

For datasets with reference images, the structural similarity (SSIM) and the peak signal to noise ratio (PSNR) are used to quantify the enhancement performance. To further demonstrate the effectiveness of the proposed restoration framework, we utilize a U-Net [27] trained on DRIVE[30] training set to segment retinal vessels in the degraded and restored DRIVE[30] testing sets, and compare the segmentation results with the manually-annotated ground truth. The segmentation accuracy is evaluated by the intersection over union (IoU) and the Dice coefficient between the restored images and the reference.

### 4.3 Comparison With State-of-the-Art

Tab. 1 compares the enhancement performance between MSANet and the state-of-the-art methods, including traditional method (SGRIF [3]), methods without using any prior (StillGAN[22], I-SECRET [4] and RFormer [6]), methods based on semantic prior (CofeNet [29]), and methods based on structure prior (PCE-Net [20]).

Compared with the traditional method SGRIF [3], our method achieves significantly better performance on downstream vessel segmentation task. Specifically, our method achieves 9.64% in IoU and 9.68% in Dice. SGRIF is a hand-crafted method tailored for cataract-affected fundus images restoration, making it prone to adjust image contrast but difficult in fragile vessel preservation.

I-SECRET [4] and StillGAN[22] are both based on GAN [11]. They utilize discriminator to align restored image with high-quality image, enabling them to generalize well to real datasets such as RF [6]. However, from I-SECRET [4] results on downstream segmentation task, it is evident that only introducing an additional decoder to assess the importance of each pixel without any prior information or domain knowledge cannot effectively preserve vessel structures.

CofeNet [29] and PCE-Net [20] respectively help the enhancement network preserve vessels by introducing semantic priors and structural priors. However, our method still improves PSNR by up to 6.203db and SSIM by up to 0.0536 in full-reference evaluation and IoU by up to 1.31% in segmentation evaluation. On one hand, these approaches [20, 29] overlook the slender and tortuous

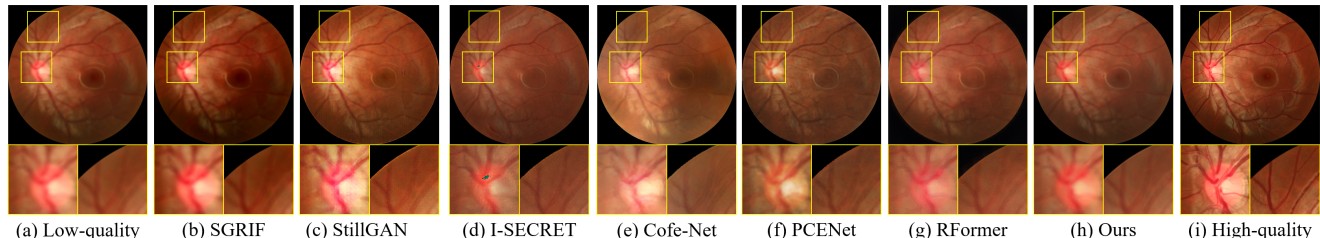

(a) Low-quality    (b) SGRIF    (c) StillGAN    (d) I-SECRET    (e) Cofe-Net    (f) PCENet    (g) RFormer    (h) Ours    (i) High-quality

**Figure 6: Fundus image enhancement on RF [6] dataset.**

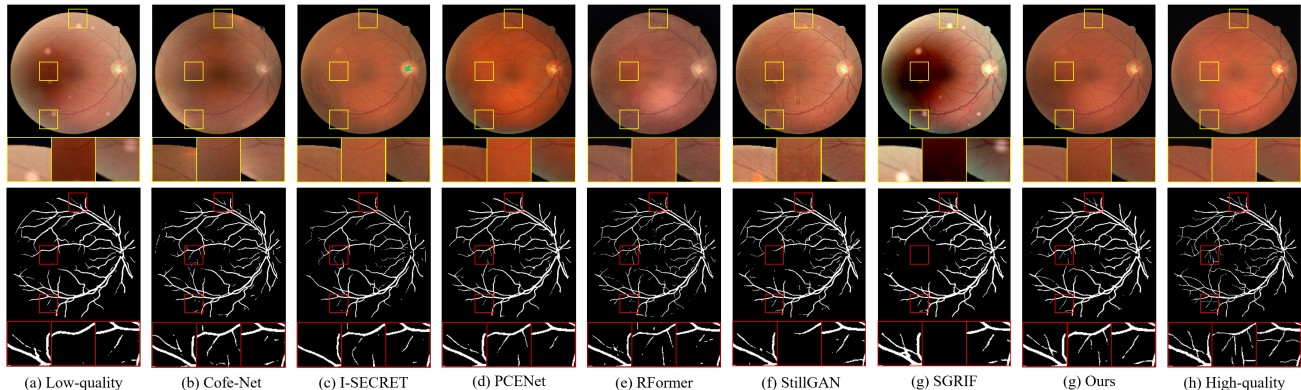

(a) Low-quality    (b) Cofe-Net    (c) I-SECRET    (d) PCENet    (e) RFormer    (f) StillGAN    (g) SGRIF    (g) Ours    (h) High-quality

**Figure 7: Fundus image enhancement on DRIVE[30] dataset. From top to bottom, the images are the restored images and their zoom-in, and the vessel segmentation results and their zoom-in. All of the segmentation results are provided by a pretrained U-Net [27].**

**Table 2: Ablation study for modules.**

| Method | Full reference | | Segmentation | |
|---|---|---|---|---|
| | PSNR | SSIM | IoU | Dice |
| Baseline | 25.0411 | 0.7874 | 0.5280 | 0.6881 |
| Baseline + SA | 25.5480 | 0.8055 | 0.5407 | 0.7000 |
| Baseline+MSA(MSANet) | **25.7261** | **0.8068** | **0.5477** | **0.7064** |

structural characteristics of retinal vessels. On the other hand, they directly concatenate prior information into the network limiting the potential of the prior information.

RFormer [6] is a Transformer-based method which uses window-based multi-head self-attention to establish dependencies between pixels without prior information guidance or domain knowledge. Therefore, our MSANet can achieve better performance than RFormer on all evaluation metrics.

In both full-reference enhancement and segmentation evaluation, our proposed MSANet method consistently outperforms these state-of-the-art methods. Fig. 6 and Fig. 7 respectively provide visual results of various methods in full-reference and segmentation evaluations on RF [6] dataset and DRIVE [30] dataset.

## 4.4 Ablation Study

To validate the effectiveness of Snake Attention and Masked Snake Attention, we conduct ablation study, including Baseline: The baseline model of MSANet by removing Snake Attention (SA) and Masked Snake Attention (MSA).

Baseline+Snake Attention(Baseline+SA): A model in which we integrate the SA module with three scales into the decoder of Base. Comparing the results of Baseline in Tab. 2, Baseline+SA achieves noticeable improvements in terms of SSIM, IoU and Dice. Especially for PSNR and IoU, Baseline+SA is much higher than Base by 0.5069db and 1.27%. It demonstrates that Snake Attention module can help the network capture the key features of thin and long vessel structures from different scales.

Baseline+Masked snake Attention(Baseline+MSA): According to Tab. 2 and Fig. 8, jointly using SA and MSA achieves the best overall performance, outperforming Baseline by up to 0.6850db in PSNR and 1.97% in IoU. It shows that MSA can perform feature aggregation more effectively and further improve performance on full reference evaluation and segmentation than SA at the top layer of MSANet.

## 4.5 Analysis of threshold

We evaluate the performance of our MSANet on RF [6] dataset at different threshold, as shown in Fig. 9. According to Eq.12 and Eq.13,

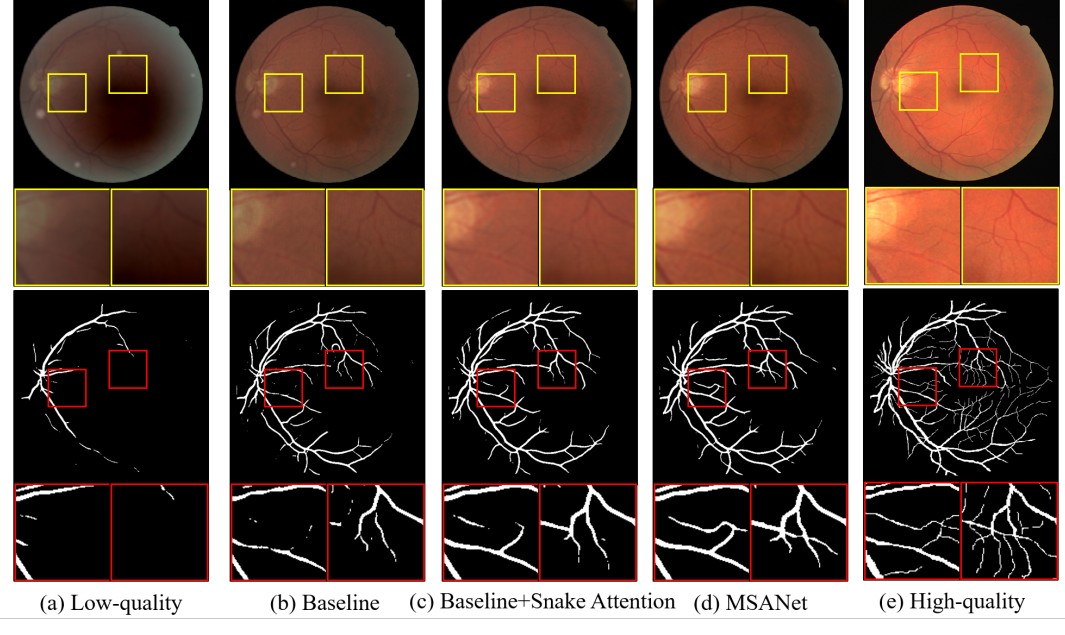

(a) Low-quality    (b) Baseline    (c) Baseline+Snake Attention    (d) MSANet    (e) High-quality

**Figure 8: Ablation study for modules on DRIVE[30] dataset. From top to bottom, the images are the restored images and their zoom-in, and the vessel segmentation results and their zoom-in. All of the segmentation results are provided by a pretrained U-Net [27].**

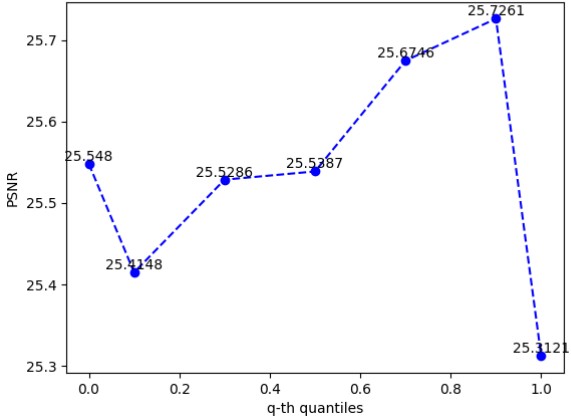

**Figure 9: Comparison of MSANet on RF [6] dataset across different thresholds.**

*threshold* decides the region which is applied with snake attention. We compute the q-th quantiles of $\mathbf{M}$ as *threshold*, where $q$ takes values of 0.0, 0.1, 0.3, 0.5, 0.7, 0.9, and 1.0. From Fig. 9, it can be observed that when the value of $q$ is too small, only a small portion of vessel pixels are enhanced through snake attention, providing limited assistance to the network in restoring fundus images. As the value of $q$ gradually increases, more pixels are enhanced through snake attention, improving the network's ability to restore fundus

images. When $q$ is set to 0.9, the gain of Masked Snake Attention (MSA) is maximized. However, when $q$ is set to 1.0, meaning $\mathcal{M} = -\infty$, MSA loses its ability to model element relations.

## 5 CONCLUSION

We propose a novel method Masked Snake Attention Network (MSANet) for accurately preseving vessel while restoring low-quality fundus images. The cores of MSANet are one Masked Snake Attention module(MSA) and two Snake Attention(SA) modules which can adaptively capture features under the domain knowledge of tubular structure morphology. Compared with SA, MSA can apply snake attention on region guided by semantic prior to more efficiently capture vessel features. Extensive experimental results demonstrate that our MSANet consistently outperforms state-of-the-art methods in terms of overall image quality and the performance in downstream segmentation tasks.

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
