# OpenReview forum: "Masked Snake Attention for Fundus Image Restoration with Vessel Preservation"
_acmmm.org/ACMMM/2024/Conference — MM2024 Poster_

### Official Review · Reviewer_aWM1 · 2024-05-14

**Rating:** 5
**Confidence:** 3

**Summary:**

In this study, the authors proposed a Masked Snake Attention module for fundus image vessel segmentation.

**Strengths:**

The Snake Attention is used to adaptively aggregate vessel features, the Masked Snake Attention is used to restrict the snake attention on localized features. There are novelty in method.

**Limitations:**

The experiment results on segmentation are not improved much.
The definition of variable ‘z’ in equation (3) is confused.

**Suitability:**

3

---

### Official Review · Reviewer_bVFm · 2024-05-20

**Rating:** 4
**Confidence:** 3

**Summary:**

A novel method Masked Snake Attention Network (MSANet) was proposed for low-quality fundus image restoration while retaining retinal vessel. The MSANet introduces Snake Attention (SA) module and Masked Snake Attention (MSA), which adaptively aggregates vessel features within the predicted vessel region based on their morphological structure.

**Strengths:**

1. The paper has a clear structure, informative content, exquisite figures, and is well written.
2. Proposed MSA and SA modules to adaptively capture features under the guidance of domain knowledge of vessel structure morphology.
3. The quantitative results of image reconstruction and vessel segmentation are superior to the methods used for comparison.

**Limitations:**

1. The effectiveness of MSANet relies on the quality of the prior information from segmentation methods. Inaccuracies in the segmentation masks could potentially affect the restoration outcome.
2. Experimental results in Table 1 were conducted on which dataset? This should be indicated. Based on description in Sec. 4.1, I guess "full reference" represents the reconstruction results on RF dataset, and "segmentation" is the segmentation results on DRIVE dataset. However, segmentation is a downstream task here, thus the reconstruction results on DRIVE dataset should also be presented.
3. The qualitative superiority of this method is not obvious in Figure 6.

**Suitability:**

2

---

### Official Review · Reviewer_kTaM · 2024-05-24

**Rating:** 4
**Confidence:** 3

**Summary:**

The authors propose a fundus image restoration method which could accurately restoring vessel structures. Snake Attention (SA) is applied to adaptively aggregate vessel features based on the morphological structure of the vessels. And the Masked Snake Attention (MSA) further restricts the snake attention on localized features centered around predicted segments.
Experiments on three public datasets demonstrate that MSANet not only enhances fundus images but also benefits the downstream segmentation tasks, and the performance was better than existing methods.

**Strengths:**

The proposed method could preserve more retinal vessel structure details than other existing image restoration methods. Those vessel structure characteristics serve as important biomarkers for many retinal and systemic diseases.

**Limitations:**

1. Missing some necessary experiments, e.g., The training and running speed of the proposed method compared with other compared methods
2. Some details about the proposed method, the training details are missing

**Suitability:**

3

---

### Meta-Review · Area_Chair_9q3s · 2024-07-09

**Recommendation:** Accept (Poster)
**Confidence:** 4

**Metareview:**

The paper introduces MSANet, a method for enhancing low-quality fundus images while preserving retinal vessel structures using Snake Attention (SA) and Masked Snake Attention (MSA). Reviewers generally appreciate the clarity, structure, and potential contributions of the proposed method in improving both image quality and downstream segmentation tasks. However, the paper should undergo revisions to improve clarity in experimental methodologies, specifically addressing the datasets used, enhancing explanations of variable definitions, and discussing the implications of segmentation accuracy on the proposed method's efficacy.